# DanTok: Domain Beats Language for Danish Social Media POS Tagging

**Kia Kirstein Hansen, Maria Barrett, Max Müller-Eberstein, Cathrine Damgaard,**
**Trine Naja Eriksen, Rob van der Goot**

IT University of Copenhagen

`[kiah, mbarrett, mamy, catd, trer, robv]@itu.dk`

## Abstract

Language from social media remains challenging to process automatically, especially for non-English languages. In this work, we introduce the first linguistically annotated dataset for TikTok comments and the first Danish social media dataset with part-of-speech annotation. Additionally, we supply annotations for normalization, code-switching, and annotator uncertainty. As transferring models to such a highly specialized domain is non-trivial, we conduct an extensive study into which source data and modeling decisions most impact the performance. Surprisingly, transferring from in-domain data, even from a different language, outperforms in-language, out-of-domain training. These benefits nonetheless rely on the underlying language models having been at least partially pre-trained on data from the target language. Using our additional annotation layers, we analyze how normalization, code-switching, and human uncertainty affect the tagging accuracy.

## 1 Introduction

Language data from social media offer unique insights into how communities use language to communicate in a natural, spontaneous setting, using a highly domain-specific vocabulary. This domain is, however, also subject to frequent changes, high noise, and variability, making it difficult to process (Eisenstein, 2013) both for high (Gimpel et al., 2011; Derczynski et al., 2013) and especially for lower-resourced languages (Kaji and Kitsuregawa, 2014; Albogamy and Ramasy, 2015; Singh et al., 2018; Mæhlum et al., 2022). To better understand and improve how to target such

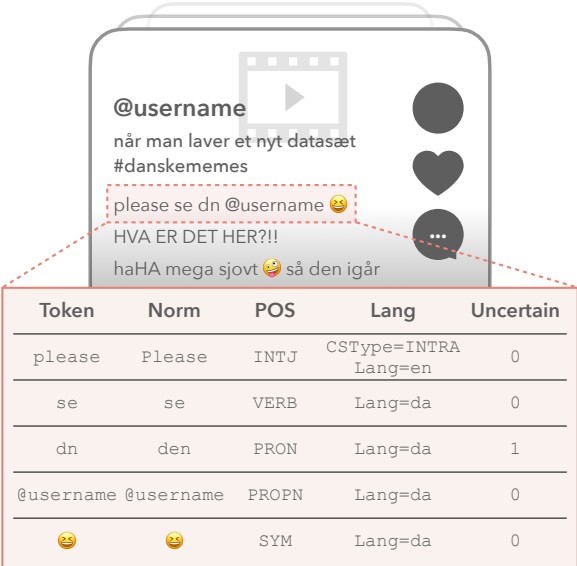

Figure 1: DanTok annotation layers for a fabricated example comment.

a highly specialized domain, we present the first linguistically annotated dataset of contemporary Danish social media from TikTok, a relatively new platform focused on short videos. The Danish Dependency Treebank (Kromann et al., 2003) from the PAROLE Corpus (Bilgram and Keson, 1998), and its Universal Dependencies (UD) conversion (Johannsen et al., 2015) is the primary part-of-speech (POS) resource for Danish (Kirkedal et al., 2019). However, it contains well-edited text only.

We present not only the first Danish social media dataset with POS annotation, but, to the best of our knowledge, also the first linguistic exploration of TikTok language in any language. The dataset contains annotation for POS, normalization, code-switching and annotator uncertainty (Figure 1). It further prioritizes quality over quantity, covering over 8k tokens, which have been manually verified to ensure relevance, correctness, and privacy.

It has been shown that training language models on the target domain can be used for cross-domain

---

Code available at: github.com/kkirsteinhansen/dantok.
Data is available upon request (contact `robv@itu.dk`).

learning (Gururangan et al., 2020; Barbieri et al., 2022). In this work, we aim to tease apart the effect of the pre-training data of the language model and the source of the annotated training data for the target task. For this purpose, we contribute: ① DanTok, the first linguistically annotated dataset of TikTok comments and the first Danish POS-annotated social media dataset; ② An extensive study of transfer learning targeting this highly specialized domain; ③ An in-depth analysis investigating the unique features of this new domain, and how specific data/model properties help boost transfer performance.

## 2 DanTok

### 2.1 Collection

The social media platform TikTok shows a continuous, personalized feed of short videos on various topics. Users may respond to these videos using likes and/or comments. In 2022, the number of Danish users of age $\geq 18$ was estimated to be 1.14M. This is substantial compared to the total population in Denmark (5.8M), as well as, e.g., the number of Danish Twitter users (est. 685k).[1] It is challenging to gather data from TikTok in a reproducible way due to algorithmically populated user feeds as well as the inability to filter videos by language. To increase reproducibility, we browsed videos directly by hashtag while logged out. We selected 15 Danish hashtags (Appendix B.1) where videos were determined to be predominantly in Danish.

**Filtering and Deduplication** We applied automatic filtering to maximize data diversity relative to the dataset size while reducing annotation workload. Using the Aspell dictionary for Danish (Aspell, 2019), we removed non-Danish comments while retaining interesting token variations using a removal threshold of $< 10\%$ or $> 60\%$ OOV (out-of-vocabulary) tokens per comment. As comments often have a high degree of repetitiveness (e.g., "wow", "woow!!"), we deduplicated them by iteratively merging the two most similar comments, keeping only one, until there were no more pairs with a similarity greater than a threshold $t$. Similarity was measured using the normalized Levenshtein distance, and we retained the comment with a higher total similarity to all other com-

ments (i.e., the more prototypical one). To determine the best threshold, we conducted a sweep over $t \in (0.0; 1.0)$ in 0.05 increments (see also Appendix B.2), and chose $t = 0.60$, as this set retained a relatively high diversity while accounting for the downstream, manual filtering of irrelevant comments down to our target of annotating 8k tokens.

### 2.2 Annotation

The data was tokenized using the `NLTK Tweet-Tokenizer` (Bird et al., 2009) with manual post-correction. We replaced usernames with *@user-name*, and manually filtered out comments containing personal information. After two rounds of annotation[2], we reached a Fleiss' $\kappa$ score of 96.05. As the remaining disagreements were mainly ambiguities and accidental mistakes, we annotated the remainder of the dataset once, with the following layers:

**POS** We followed the UD 2.11 guidelines and the social media-specific suggestions by Sanguinetti et al. (2020). Where possible, we annotated the intended meaning of a token given its use in context, e.g., in "Would you like a 🍺?", 🍺 is a `NOUN`. Tokens that imitate pronunciation were tagged as `INTJ`. *Den*, *det* and *de*, which may function as articles or pronouns, were tagged as `DET` only when followed by an adjective.

**Normalization** Given that we annotated the intended meaning of a token, we additionally decided to annotate for lexical normalization. We followed the format and guidelines of Multi-LexNorm (van der Goot et al., 2021a): URLs and interjections were not normalized, and splits and merges were included in the annotations. Merging of tokens that had erroneously been written as multiple words is indicated in the normalization column of the last word, as compound words in Danish adopt the part-of-speech (POS) of the final word. For tokens that were split, an overall POS tag was given to this token alongside individual tags for the tokens resulting from the splitting.

**Code-Switching** Code-switching is indicated with language and type. The type is either `INTER` (full sentence), `INTRA` (individual tokens) or `MIXED` (a lemma from one language with inflection from another language) as described by

---

[1] https://datareportal.com/reports/digital-2022-denmark

[2] See Appendix A for information about the annotators.

| | Langs | Train-sources | Type | Train-size | Architecture | Params | %Unk. | Subw. ratio | Reference |
|---|---|---|---|---|---|---|---|---|---|
| **LLM** | | | | | | | | | |
| DANISH-BERT-botxo | DA | web, wiki, subtitles | | 10gb | Bert-base | 111M | 0.15 | 1.28 | github.com/certainlyio/nordic_bert |
| RØBÆRTA-base-danish | DA | web | -D+L | ? | Roberta-base | 125M | 0.00 | 1.58 | hf.co/DDSC/roberta-base-danish |
| ÆLÆCTRA-danish-small-cased | DA | legal, social, web, wiki, news | | 1,045M words | Electra-small | 14M | 0.04 | 1.39 | github.com/MalteHB/-l-ctra |
| BERTWEET-Base | EN | social | | 850M tweets | Roberta-base | 135M | 0.10 | 1.65 | Nguyen et al. (2020) |
| BERTWEET-Large | EN | social | +D-L | 850M tweets | Roberta-large | 355M | 0.00 | 1.90 | Nguyen et al. (2020) |
| TWITTER-ROBerta-base | EN | social | | 58M tweets | Roberta-base | 125M | 0.00 | 1.90 | Barbieri et al. (2020) |
| TWITTER-XLM-roberta-base | 30+ | social | | 198M tweets | XLM-r base | 278M | 0.01 | 1.45 | Barbieri et al. (2022) |
| BERNICE | 66 | social | +D-L | 2.5B tweets | Roberta-base | 278M | 0.00 | 1.44 | DeLucia et al. (2022) |
| TWHIN-bert-large | 100+ | social | | 7.5B tweets | new | 561M | 0.01 | 1.45 | Zhang et al. (2022) |
| **TREEBANK** | | | | | | | | | |
| LINES | EN | fiction, nonfiction, spoken | -D-L | 57,372 words | | | | | Ahrenberg (2015) |
| TWEEBANK2 | EN | social | +D-L | 24,753 words | | | | | Liu et al. (2018) |
| DDT | DA | fiction, nonfiction, spoken, news | -D+L | 80,378 words | | | | | Johannsen et al. (2015) |

Table 1: An overview of the used language models and POS fine-tuning sets. %Unk. is the percentage of unknown subwords in our development data; Subw. ratio is the average amount of subwords per word. Capitalized name parts are handles.

| | COMMENTS | TOKENS | TYPES | TTR |
|---|---|---|---|---|
| Dev | 429 | 4,000 | 1,520 | 0.38 |
| Test | 430 | 4,028 | 1,519 | 0.38 |
| **Total** | 859 | 8,028 | 2,512 | 0.31 |

Table 2: DanTok dataset statistics. TTR is type-token ratio.

Sanguinetti et al. (2020). We used the Danish dictionary[3] for cross-referencing, as many originally English words are now considered Danish.

**Certainty** Following Bassignana and Plank (2022), the annotator's certainty of a POS tag was annotated as either 0 (certain) or 1 (uncertain).

### 2.3 DanTok Statistics

Our final dataset consists of 8,028 tokens and 2,512 unique types (Table 2). A comparative POS tag distribution is given in Appendix C. In Dan-Tok, we observe that 16.66% of the tokens required normalization, 5.03% were code-switched (all to English), and 5.12% had annotation uncertainty. Overall, these annotation layers allow us to investigate how Danish is used on contemporary internet platforms with respect to syntax, and how sociolinguistic factors such as code-switching can impact downstream performance.

## 3 Experiments

### 3.1 Setup

For a highly specialized dataset such as DanTok, transfer learning is key, as there is no training data matching the domain and language. We therefore investigated 36 combinations of in/out-of-domain (+D/-D)[4] and in/out-of-language (+L/-L)[5] training data and large language models (LLMs). We selected English as the -L transfer language due to dataset and language model availability. All experiments were replicated on the normalized version of DanTok. The Danish LLMs are trained on web data, including some forum data, but none are explicitly optimized for social media. The LLMs and training sets used in our experiments are given in Table 1. All the models consist of an LLM encoder plus a linear layer for POS labeling (both fully finetuned) and are implemented in MaChAmp v0.4 (van der Goot et al., 2021b) using default hyperparameters with the development data for model selection. To avoid overfitting on DanTok, we use the transfer data's development set for model selection (Artetxe et al., 2020).

### 3.2 Results

Our main results are given in Table 3. Unsurprisingly, the combination of in-domain, in-language (+D+L) training data and LLMs results in the best overall performance. In general, having in-language data is more beneficial than in-domain data; however, when training on a single dataset, the in-domain English dataset (+D-L) leads to surprisingly high performance with the multilingual language models, even outperforming all scores obtained with the Danish training data (-D+L). One reason for this could be the relatively high frequency of code-switched tokens (5%). Inter-

---

[3] https://dsn.dk/ordboeger/retskrivningsordbogen/

[4] +D: social media data, -D: data from other domains.

[5] +L: trained on Danish, -L: trained on other languages.

| | DATA | -D-L | +D-L | -D+L | +D-L + -D+L |
|---|---|---|---|---|---|
| MODEL | | LINES | TWB | DDT | TWB+DDT |
| **-D+L** | DANISH-BERT | 44.02 | 49.60 | 77.98 | 84.08 |
| | RØBÆRTA | 58.43 | 60.82 | 70.17 | 78.72 |
| | ÆLÆCTRA | 49.50 | 63.30 | 74.20 | 84.95 |
| **+D-L** | BERTWEET-B | 27.80 | 38.00 | 67.90 | 79.47 |
| | BERTWEET-L | 25.92 | 36.55 | 67.40 | 81.50 |
| | TWITTER-ROB | 25.02 | 37.30 | 64.05 | 79.40 |
| **+D+L** | TWITTER-XLM | 67.58 | 77.15 | 72.15 | 83.28 |
| | BERNICE | 70.45 | 78.22 | 72.95 | 83.28 |
| | TWHIN | 69.30 | 81.38 | 72.65 | **85.92** |

Table 3: POS tagging accuracy on the DanTok development set using combinations of in/out-of-domain (+D/-D) and in/out-of-language (+L/-L) models and training data, plus a concatenation covering +D and +L.

| | DATA | -D-L | +D-L | -D+L | +D-L + -D+L |
|---|---|---|---|---|---|
| MODEL | | LINES | TWB | DDT | TWB+DDT |
| **-D+L** | DANISH-BERT | 42.69 | 48.12 | 80.19 | 85.75 |
| | RØBÆRTA | 60.79 | 63.48 | 73.80 | 82.36 |
| | ÆLÆCTRA | 50.87 | 61.69 | 78.48 | 88.45 |
| **+D-L** | BERTWEET-B | 28.24 | 38.48 | 70.88 | 82.83 |
| | BERTWEET-L | 27.46 | 37.48 | 70.98 | 85.35 |
| | TWITTER-ROB | 25.75 | 38.13 | 68.76 | 84.19 |
| **+D+L** | TWITTER-XLM | 69.85 | 80.12 | 75.16 | 86.26 |
| | BERNICE | 72.01 | 80.27 | 75.61 | 85.15 |
| | TWHIN | 70.95 | 83.09 | 75.33 | **88.80** |

Table 4: POS tagging accuracy on the normalized DanTok development set using combinations of in/out-of-domain (+D/-D) and in/out-of-language (+L/-L) models and training data.

estingly, model size (see Table 1) is not a good predictor of performance: Although the largest model, TWHIN, obtains the highest score overall, it requires large amounts of pre-training data and a specialized pre-training objective based on rich social engagements (Zhang et al., 2022). Meanwhile, ÆLÆCTRA's performance is very close, despite being 41 times smaller. Given these results, we conclude that the best strategy for obtaining a high-quality tagger would be to use domain-specific models when available (even if multilingual) and use in-domain fine-tuning data even if in another language (+ in-language if available).

Table 4 shows that using normalized data gives a consistent boost of 2–5 % points across all setups, with only a few exceptions. Furthermore, performance varies less compared to the non-normalized data (Table 3).

| LLM | -NORM | +NORM |
|---|---|---|
| TWHIN | 86.05 | 88.18 |
| ÆLÆCTRA | 85.80 | 88.55 |

Table 5: Results on the DanTok test set of our two best models trained on TWEEBANK and DDT.

**On Test Data** TWHIN performs similarly on the development and test data. After normalization, the smaller ÆLÆCTRA model outperforms TWHIN slightly (Table 5).

## 4 Analysis

### 4.1 Subword Analysis

The Subword ratio (Table 1) does not show a clear correlation with performance, so we qualitatively evaluate the subword segmentation of the two best-performing models, TWHIN and ÆLÆC-TRA. Surprisingly, we find that the multilingual model (TWHIN) seems more capable of interpreting inflection suffixes than the Danish model. It correctly splits morphemes indicating definiteness, plurality, or adverbial status, which the Danish model sometimes fails to do. Examples of this are *batterier* ("batteries") split into *batteri-er* ("batteri-es") and *dårligt* ("badly") split into *dårlig-t* ("bad-ly") only by the multilingual model, whereas the Danish model does not split these tokens at all.

### 4.2 Stratified Analysis

We explore the accuracy on different subsets of the development set according to our additional annotation layers (Table 6). We observe that the models, perhaps unsurprisingly, struggle more with tokens that were normalized, as well as tokens that annotators were also uncertain of. For code-switched tokens, we observe a large performance drop for the in-language LLM (ÆLÆCTRA) despite fine-tuning on English in-domain data. Surprisingly, the multilingual model, likewise fine-tuned on Danish and English in-domain data, also struggles with code-switched tokens.

### 4.3 Qualitative Error Analysis

The most frequent tag confusions for the best ÆLÆCTRA model are given in Figure 2. TWHIN follows a similar pattern. Over half of the errors made by each tagger on the original data are shared with the other tagger. Some of the errors

| LLM | POS CERTAINTY | | | NORMALIZED | | | IN FINE-TUNE VOCAB | | | CODE-SWITCHED | | |
|---|---|---|---|---|---|---|---|---|---|---|---|---|
| | $n$ | - | + | $n$ | - | + | $n$ | - | + | $n$ | - | + |
| TWHIN | 203 | 62.1 | 87.2 | 3,338 | 88.9 | 70.8 | 1,234 | 83.3 | 87.1 | 3,808 | 86.3 | 78.1 |
| ÆLÆCTRA | | 58.6 | 86.4 | | 88.6 | 66.8 | | 83.1 | 85.8 | | 86.1 | 62.5 |

Table 6: Stratified accuracy on the 4,000-token dev set of the two best models trained on TWEEBANK and DDT. $n$ is the number of tokens in the - category, e.g., 1,234 words were not seen during fine-tuning.

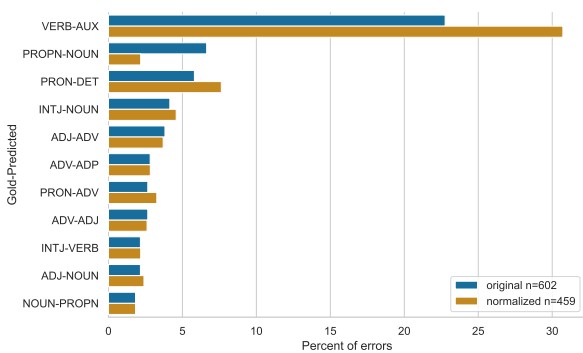

Figure 2: The 11 most frequent tag confusions for the ÆLÆCTRA model.

are caused by erroneous annotations in DanTok. The most frequent error types can be categorized as follows:

**VERB vs. AUX** In DanTok, the present tense of the copula verb *være* ("to be") has been labeled VERB when it is the only verb in the sentence. However, the models prefer the tag AUX in 91.6% and 85.0% of cases, respectively. This seems to be in line with the UPOS guidelines and is likely a result of the annotation of *er* in the DDT training set; here, 78.85% of *er* tokens have been tagged as AUX (the remaining being tagged as VERB).

**Pronoun Confusions** Tokens that may be multiple parts of speech confuse the taggers. The most frequent issue is PRON and DET confusion, which is arguably non-trivial in Danish[6]. PRON and ADV confusion is also prevalent; e.g., the token *der* can be either a relative PRON, the preliminary subject "there", or an ADV of place. In the erroneous predictions, *der* is generally tagged as ADV.

**Proper Noun Inconsistencies** Orthographic variations in social media language throw off the models. For example, names written in lowercase are often tagged as NOUN rather than PROPN. On the normalized data, PROPN (gold) → NOUN errors decreased by 75% for ÆLÆCTRA and 62%

for TWHIN. Likewise, when capitalized names are used in context, the models labeled them as PROPN, whereas we annotated the syntactical use of the token, e.g., *filming a TikTok*/NOUN.

**ADV vs. ADP** These are errors made on tokens like *af* ("of, off") and *for* ("for, too") which may function as both prepositions and adverbs[7]. In a few cases, the models do not recognize when *for* is used as an adverb of degree.

**ADJ vs. ADV** For adjectives that end in *-t*, the models seem to prefer the ADV tag. While *-t* can indicate an adverb, it may also indicate the gender of an adjective. The token *her* ("here"), an adverb, also poses a challenge when it occurs before a noun, e.g. *den her bog* ("this book"). In such cases, the models seem to prefer the erroneous tag sequence *den*/DET *her*/**ADJ** *bog*/NOUN.

**Interjection Confusions** Tokens that are meant to imitate pronunciation have been labeled INTJ in DanTok, but the models seem to prefer a more concrete labeling[8]. The models also prefer INTJ for tokens with character repetition, whereas we tagged these tokens according to their presumed intended function.

## 5 Conclusion

We presented DanTok, the first linguistically annotated TikTok dataset and the first Danish social media dataset with POS annotation. We conducted an extensive analysis of how to best transfer to a highly specialized domain in a mid-resource language, and we demonstrated that LLMs benefit from common approaches such as normalization, while struggling with the same cases as the human annotators. Simultaneously, our results show that although in-language data and models form the basis for high performance, in-domain data, even from another language, should not be neglected in order to achieve state-of-the-art results.

---

[6]Consider, e.g., *den/**PRON** bog*/NOUN vs. *den/**DET** gamle*/ADJ *bog*/NOUN ("**that** book" vs. "**the** old book").

[7]*For* may also be used as a conjunction.

[8]E.g., "It's *nuclear*, not *nucular*," should be tagged as if it said *nuclear* twice.

## Acknowledgments

We would like to thank the members of NLP-north, MaiNLP Lab and the anonymous reviewers for their feedback. This project has been supported by the Pioneer Centre for Artificial Intelligence. Maria Barrett is supported by a research grant (34437) from VILLUM FONDEN. Max Müller-Eberstein is supported by the Danmarks Frie Forskningsfond (DFF) Sapere Aude grant 9063-00077B.

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

# Appendix

## A  Data Statement

The following dataset characteristics are outlined following Bender and Friedman (2018):

A. CURATION RATIONALE  This dataset aims to provide high-quality, linguistically annotated data from contemporary Danish social media, in order to allow for analyses of how language use is evolving in these specialized domains, and how NLP methods can best be adapted to these changes.

B. LANGUAGE VARIETY  The data consists of comments from TikTok videos collected in January 2023. The language covered is manually verified Danish (da-DK) with code-switching to English (en), and orthographic variations specific to the social media domain.

C. SPEAKER DEMOGRAPHIC  Nothing specific is known about speaker demographics, as the data was scraped from 75 videos spanning different topics.

D. ANNOTATOR DEMOGRAPHIC  Three Master's students, all native Danish speakers, one with previous experience in dataset creation for POS tagging. The annotators were paid for their efforts.

E. SPEECH SITUATION  Comments under TikTok videos represent informal, written language produced largely spontaneously with the intent to address the video creator or express an opinion to other viewers.

F. TEXT CHARACTERISTICS  The text contains domain-specific terms and abbreviations, some degree of typographical and orthographic errors as well as occasional ellipsis of sentence subject. Code-switching to English makes up 5% of tokens in the full dataset (development + test), though the dataset contains several additional tokens that exist with the same meaning in both English and Danish, e.g., *shit* and *like*.

G. RECORDING QUALITY  N/A

H. OTHER  N/A

I. PROVENANCE APPENDIX  N/A

## B  Data Collection Details

### B.1  Hashtags

Videos from the following 15 hashtags were scraped during data collection:

- #børn ("children")
- #danskememes ("Danish memes")
- #danskhumor ("Danish humor")
- #glædeligjul ("merry Christmas")
- #godtnytår ("happy new year")
- #gørdetselv ("do it yourself")
- #landsholdet ("the national team")
- #madlavning ("cooking")
- #mitarbejde ("my job")
- #morgenrutine ("morning routine")
- #parforhold ("relationships")
- #selvtak ("you're welcome")
- #sommerprojekt ("summer project")
- #tobiasrahim ("Tobias Rahim")
- #træning ("workout")

### B.2  Deduplication Details

Figure 3 plots the number of tokens and their token-type ratios (TTR) after applying merge deduplication (Section 2.1) with threshold $t$.

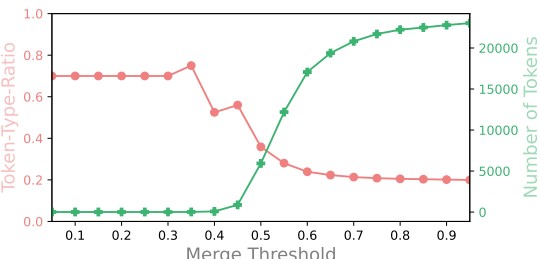

Figure 3: Deduplication using varying merge thresholds. A low $t$ merges all comments into one, a high $t$ contains more tokens and less token-type diversity.

## C  POS Tag Distribution

Figure 4 presents an overview of the POS tag distribution in DanTok compared to the English LINES, TWEEBANK2 and DDT.

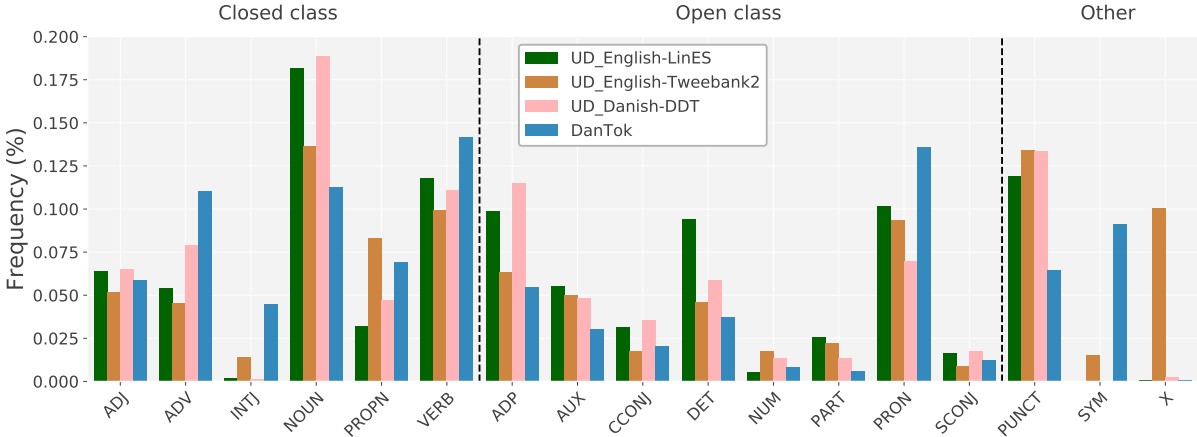

Figure 4: POS tag distribution in DanTok compared to the treebanks used for fine-tuning in Section 3.