# OpenReview forum: "DanTok: Domain Beats Language for Danish Social Media POS Tagging"
_NoDaLiDa/2023/Conference — NoDaLiDa 2023_

### Official Review · Reviewer_Ly3R · 2023-02-19

**Rating:** 9
**Confidence:** 4

**Review:**

This paper described an annotation effort of a new corpus of Danish TikTok comments, and an empirical study of the performance of POS-tagging for this data set for a variety of pre-trained models and parsing data, with different combinations of language and domain.

I think the authors did a good job describing this study, giving the limitations of a short paper. The data collection as well as the experiments are described in enough detail. If the paper is accepted I would recommend moving the more detailed results in Table 7 to the main paper.

For your UD datasets not matching language, you have chosen Swedish, which is very close to Danish, and English, which is also close in structure, and for which there is a 5% token overlap due to code-switching. I do not think these are truly not matching in language. I think you at least need a discussion of this fact in the paper. If time permits, it would be interesting to see what happens if you use languages further away (e.g. Italian, which has both Twitter data and other data in UD).

Can you motivate the choices of tagging you describe in section 2.2? Are these consistent with UD guidelines, or does it deviate? If it deviates, why did you make these choices?

Overall, I think this is a strong short paper, providing both interesting experimental results, and describing a useful corpus.





**Paper Type:**

Short paper

---

### Official Review · Reviewer_JKRu · 2023-03-06
**A good paper presenting the first POS annotated Danish social media dataset, evaluation of tagging results and error analysis.**

**Rating:** 7
**Confidence:** 4

**Review:**

This a good (student) paper presenting the first POS annotated Danish social media (TikTok) dataset.
Furthermore, it presents an evaluation of transfer learning targeting the social media domain and analysis of tagging errors.

The evaluation part is especially interesting, in which various combinations of in/out domain training data and in/out language models are considered.

The main problem with the paper is that it seems that the authors have tried to squeeze the material into four pages (for a short paper) and, as a result, the Appendix is too long and contains material that should be in the main text.  The paper should be changed to a long paper (if the program committee allows it), with minimal material in the Appendix.

Overall, the text is clear.  The following list, however, contains issues that need to be fixed.

Citations and bibliography
--------------------------
Many of the entries in the bibliography need to be checked for correct title casing. One example is:
"Universal dependencies for danish."  -> "Universal Dependencies for Danish."


Clarifications/fixing needed
----------------------------

In the Introduction, you state "... making it extremely difficult to process both for high and especially for lower-resourced languages".  You need to provide explanation for the reader why social media language data is especially difficult to process for low-resource languages.

Figure 1 is not readable enough.

"We then filtered these comments first based on % out-of-vocabulary tokens," -> "We then filtered these comments, first based on percentage of out-of-vocabulary tokens,".  What was the threshold for this filter?

If this paper is accepted, you will have an additional page for the final version.  This means that you can elaborate on some issues that you intentionally might have left out in the submitted version (due to page constraints).  In this regard, I suggest that you explain, in more detail:

	* the purpose of inter-annotator agreement convergence and the interpretaton of Fleiss score
	* why normalization is necessary and give an example of a normalizaton of a token
	* the certainty of a POS tag

Furthemore, you need to increase the font size used in Table 2.  You could skip the "Reference" column and put the references in as footnotes below the table.

In Section 3.1 Setup, you need to make clear that you are using pre-trained LMs, which you finetune on four diffrent POS datasets (the word "finetune" does not appear in that section)

It is not clear to the reader why you split DanTok into a development and test set, shown Table 1.  I assume it is because you use the development set to select the best models to be evaluated on the test data.  This shouild be specifically mentioned.

You should move Tables 5 and 6 from the Appendix into the main text.  B.2 Collection (in the Appendix) should be part of the main texdt.

"Surprisingly, we find that the multilingual model (‘TWHIN’) seems more capable of interpreting inflection suffixes than the Danish model."	-> How do you explain this?



Spelling, grammar and minor issues
----------------------------------


Introduction:
"... is the primary part-of-speech (POS) resource for Danish NLP containing only well-edited text." -> "... is the primary part-of-speech (POS) resource for Danish NLP.  However, it only contains well-edited text."

"The dataset contains annotation for POS, normalization, code-switching as well as annotation uncertainty (Figure 1)." -> "(see Figure 1)."

DanTok:
"Where possible; we annotated the intended meaning of a token ..." -> Where possible, we annotated the intended meaning of a token ..."

"For demonstrative pronouns that may also function as determiners, these were tagged as DET when followed by an adjective ..." -> "Demonstrative pronouns that may also function as determiners, were tagged as DET when followed by an adjective ..."

"Our final dataset consists of 8,028 tokens total and 2,512 unique types (Table 1)" -> "Our final dataset consists of 8,028 tokens in total and 2,512 unique types (see Table 1)"

"We observed 16.66% of tokens requiring normalization, 5.03% code-switching (all to English), and 5.12% of tokens with annotation uncertainty." ->
"We observed that 16.66% of the tokens required normalization, 5.03% code-switching (all to English), and that 5.12% of the tokens had annotation uncertainty."

Analysis:
"The Subword ratio (Table 2)" -> "The Subword ratio (see Table 2)"

**Paper Type:**

Short paper

---

### Official Review · Reviewer_vwK7 · 2023-03-10
**A Danish Tiktok corpus and tagging experiments**

**Rating:** 8
**Confidence:** 4

**Review:**

This paper describes a newly collected Danish Tiktok corpus, including experiments of transfer learning for PoS annotation of the data. The paper is generally well-written and easy to follow, with a nice analysis section.

A few comments and questions:

- Although this is the first Tiktok corpus, you may want to mention some related work, e.g. social media corpora etc for Danish, and maybe what taggers have previously been used for Danish?
- In Sect 2.1 you talk about the collection procedure. Although I understand WHAT you have done, it is not entirely clear WHY. Could you motivate why you think maximizing diversity in this way is so important?
- I think the data and the experiments are interesting and you put a lot of information into the Appendix. Maybe this should be converted into a long paper instead?

Minor language and formatting comments:
- Table 2: the text is very small, could you organize the table differently to make it readable?

**Paper Type:**

Long paper

---

### Decision · Program_Chairs · 2023-03-17

Accept